# Modulating the Performance of the SAW Strain Sensor Based on Dual-Port Resonator Using FEM Simulation

**DOI:** 10.3390/ma16083269

**Published:** 2023-04-21

**Authors:** Chunlong Cheng, Zihan Lu, Jingwen Yang, Xiaoyue Gong, Qingqing Ke

**Affiliations:** 1School of Microelectronics Science and Technology, Sun Yat-sen University, Zhuhai 519082, China; 2Guangdong Provincial Key Laboratory of Optoelectronic Information Processing Chips and Systems, Sun Yat-sen University, Zhuhai 519082, China

**Keywords:** SAW, strain sensor, dual-port resonator, finite element model, structural optimization

## Abstract

Surface acoustic wave (SAW) strain sensors fabricated on piezoelectric substrates have attracted considerable attention due to their attractive features such as passive wireless sensing ability, simple signal processing, high sensitivity, compact size and robustness. To meet the needs of various functioning situations, it is desirable to identify the factors that affect the performance of the SAW devices. In this work, we perform a simulation study on Rayleigh surface acoustic wave (RSAW) based on a stacked Al/LiNbO_3_ system. A SAW strain sensor with a dual-port resonator was modeled using multiphysics finite element model (FEM) method. While FEM has been widely used for numerical calculations of SAW devices, most of the simulation works mainly focus on SAW modes, SAW propagation characteristics and electromechanical coupling coefficients. Herein, we propose a systematic scheme via analyzing the structural parameters of SAW resonators. Evolution of RSAW eigenfrequency, insertion loss (*IL*), quality factor (*Q*) and strain transfer rate with different structural parameters are elaborated by FEM simulations. Compared with the reported experimental results, the relative errors of RSAW eigenfrequency and *IL* are about 3% and 16.3%, respectively, and the absolute errors are 5.8 MHz and 1.63 dB (the corresponding *V_out_*/*V_in_* is only 6.6%). After structural optimization, the obtained resonator *Q* increases by 15%, *IL* decreases by 34.6% and the strain transfer rate increases by 2.4%. This work provides a systematic and reliable solution for the structural optimization of dual-port SAW resonators.

## 1. Introduction

Real-time strain monitoring has wide applications in industrial manufacture, civil infrastructure, motor industries, aerospace and satellite communication for structural health monitoring and failure prevention [1,2,3]. Surface acoustic wave (SAW) resonators are intensively studied and hold the promise of in situ applications of temperature [4,5], torque [6,7], strain [8,9], etc., owing to their attractive features, such as passive wireless sensing ability, simple signal processing, high sensitivity, compact size and robustness. For example, various SAW sensors with tailored designs have been utilized to measure physical quantities, including temperature [10,11], torque [12,13], strain [14,15] or chemical/biological mass loading [16,17]. The insertion loss, sensitivity and resolution of SAW sensors are closely related to the geometries of device and characteristics of material properties. Therefore, it is desirable to model and simulate a SAW device before fabrication in order to promote the performance, allowing the optimization of the SAW parameters.

The finite element model (FEM), well implemented by some packages such as COMSOL, is widely used for numerically simulating SAW devices. Zaid. T. Salim et al. [18] constructed a three-dimensional (3D) FEM of dual-port layered SAW devices, and analyzed its frequency response and electromechanical coupling coefficient with different piezoelectric layer thicknesses. Honglang Li et al. [19] obtained the propagation characteristics of RSAW on piezoelectric substrates by using the time-domain analysis method based on a 3D FEM. However, commercial FEM based on COMSOL software requires a considerable amount of computation time and a large amount of random access memory to simulate a full-sized SAW device even in two-dimensional (2D) approximation [20]. Nabila Belkhelfa and Rafik Serhane [21] used 2D periodic FEM to simulate a Rayleigh wave based on stacked Al/AlN/Si (100) devices. Evolution curves with respect to acoustic phase velocity, reflectivity and electromechanical coupling coefficient for different Al electrode patterns and different piezoelectric AlN layer thicknesses were quantified. J. Koskela et al. [22] and Zhenglin Chen et al. [23] used hierarchical cascading technology to accelerate the simulation of a full-sized 2D FEM with periodic structure. Most of these simulation studies mainly focused on SAW modes, SAW propagation characteristics and electromechanical coupling coefficients. However, there is limited work that evaluates or investigates the performance of SAW sensors with respect to insertion loss (*IL*), quality factor (*Q*) and strain transfer characteristics, which actually plays a dominant role in determining the performance of SAW strain sensor.

In this work, we systematically simulated and studied the SAW strain sensor based on a dual-port resonator. The COMSOL Multiphysics 5.6 commercial finite element software was used to establish 2D models for a quantitative analysis of the total displacement field of RSAW, eigenfrequency and frequency domain, as well as a 3D model for revealing behaviors of strain transfer. Aiming to optimize the performance of a sensor with respect to the *IL*, *Q* and strain transfer characteristics, we systematically investigated and evaluated the effects of the structural parameters of the dual-port resonator, including the number of input/output interdigital transducer (IDT) finger pairs (*N_t_*), the number of reflection grating (RG) pairs (*N_r_*), metallization rate (*η*), electrode thickness (*th_Al_*), acoustic aperture (*W*), the spacing between input and output IDTs (*L_tt_*), the spacing between RGs and IDTs (*L_rt_*) and the thickness of the piezoelectric substrate (*th_LN_*). This result helps to identify the key factors that govern the performance of the SAW devices. 

## 2. Theoretical Basis

As shown in Figure 1, a dual-port SAW resonator is usually composed of a piezoelectric substrate, input/output IDTs deposited on the surface of the piezoelectric substrate and RGs on the two ends. The IDTs can directly excite and receive SAW. When an input electrical signal is applied at the input end, it is converted into mechanical energy through the inverse piezoelectric effect and propagates on the surface of the piezoelectric substrate in the form of SAW. When the SAW signal reaches the output IDTs, it is again converted into an output electrical signal by the piezoelectric effect of the substrate, and accordingly the sensing function is realized by identifying the change of the resonator eigenfrequency. 

The propagation of SAW in a piezoelectric material is governed by the continuum equations of motion, Maxwell’s equations under the quasi-static assumption, the strain-mechanical displacement relations, the piezoelectric constitutive relations, and the appropriate boundary conditions [24]. The piezoelectric constitutive relations in stress-charge form [25] are (to simplify the expressions, the Einstein summation convention is used in the full text):(1)$$T_{\mathrm{ij}}=C_{ijkl}^{E}\cdot S_{\mathrm{kl}}-e_{ikl}^{T}\text{·}E_{k}$$
(2)$$D_{i}=e_{\mathrm{ikl}}\cdot S_{\mathrm{kl}}+\varepsilon_{ik}^{S}\text{·}E_{k}$$
where, *T_ij_* and *D_i_* represent, respectively, the second-order stress tensor and the electrical displacement vector (C/m^2^). *E_k_* is the electric field vector (V/m), *S_kl_* is the second-order strain tensor. *C_ijkl_*, *e_ikl_* and *ε_ik_* are the fourth-order elasticity tensor (N/m^2^), third-order piezoelectric tensor (C/m^2^) and second-order permittivity tensor (F/m), respectively, which can be represented as 6 × 6 matrix *C_E_*, 6 × 3 matrix *e* and 3 × 3 matrix *ε_s_*. The mechanical behavior of linear elastic materials is governed by the equation of motion:(3)$$\rho \frac{\partial^{2}u_{i}}{\partial t^{2}}=\frac{\partial T_{ij}}{\partial x_{j}}(i=1,2,3)$$
where *ρ* is the density of the material and *u_i_* is the global displacement.

Generally, the propagation velocity of a SAW is 4~5 orders of magnitude lower than that of an electromagnetic wave. Therefore, the electromagnetic field coupled with SAW can be approximated as an electrostatic field, and $E_{k}$
can be expressed as the gradient of a potential function Ф:(4)$$E_{k}=-\frac{\partial Ф}{\partial x_{k}}(k=1,2,3)$$

The relationship between strain and displacement in piezoelectric materials is:(5)$$S_{kl}=\frac{1}{2}(\frac{\partial u_{k}}{\partial x_{l}}+\frac{\partial u_{l}}{\partial x_{k}})\left(k,l=1,2,3\right)$$

Since the medium is an insulator and there is no free charge, the divergence of the electric displacement vector *D* must be equal to zero:(6)$$\frac{\partial D_{i}}{\partial x_{i}}=0$$

Using the above six equations, the wave equations can be established as [17]:(7)$$C_{ijkl}^{E}\frac{\partial^{2}u_{l}}{\partial x_{j}\partial x_{k}}+e_{ikl}\frac{\partial^{2}Ф}{\partial x_{l}\partial x_{k}}=\rho \frac{\partial^{2}u_{i}}{\partial t^{2}}$$
(8)$$e_{ikl}\frac{\partial^{2}u_{l}}{\partial x_{i}\partial x_{k}}-\varepsilon_{ik}^{s}\frac{\partial^{2}Ф}{\partial x_{i}\partial x_{k}}=0$$
for *i, j, k, l* = (1, 2, 3).

Equations (7) and (8) can be used to calculate the wave velocity, displacement and voltage at each node once the boundary conditions are set and the discretization using the finite element method is performed [26]. Therefore, it is convenient to obtain the input and output voltage. The eigenfrequency $f_{0}$ of the SAW resonator mainly depends on SAW wavelength *λ* and the phase velocity *ν* of the SAW in the piezoelectric material:(9)$$f_{0}=\frac{\nu }{\lambda }$$

The *IL* of the device represents the energy utilization efficiency, which is found by taking the frequency domain analysis and is defined as [27]: (10)$$IL\left(dB\right)=-20{log}_{10}(\left|\frac{V_{out}}{V_{in}}\right|)=-S_{21}$$
where *V_in_* is the voltage on the input IDT and *V_out_* is the voltage on the output IDT. Normally, *IL* can be characterized by *−S*_21_.

*Q* determines the sensor resolution and wireless transmission distance [28,29]. It is generally defined as the ratio of the IDTs’ average energy storage to power loss at eigenfrequency $f_{0}$, where the average energy storage and power loss are calculated by the finite element simulation software:(11)$$Q=2\pi f_{0}\frac{Energy\;Stored}{Power\;Loss}$$

## 3. Simulation Setups

For a frequency domain analysis, a reasonable setting of the scanning range of frequency can effectively reduce the computational load. Therefore, we first establish a simplified 2D model (Figure 2a) for eigenfrequency analysis to obtain the approximate eigenfrequency of RSAW. 

However, for dual-port SAW resonators, the structural parameters mentioned in the introduction section and the profile of the total displacement of RSAW amplitude over the resonator cannot be presented in a simplified 2D model. Therefore, it is necessary to establish a full-sized 2D model to analyze the distribution of the total displacement field of RSAW and conduct a frequency scanning near the eigenfrequency of RSAW to analyze the influences of these structural parameters on the performances of the dual-port SAW resonator, as shown in Figure 2b. The accuracy of FEM simulation depends on the number of mesh elements. Generally, in one unit of size, the more mesh elements there are, the higher the accuracy will be, but the computational load increases as well. Since the structural optimization of a full-sized 2D model is time consuming, before conducting the frequency domain analysis of the full-sized 2D model, we first explore the means of meshing to reduce the calculation quantity as much as possible on the premise of ensuring accuracy.

Moreover, the characteristics of strain transfer from the tested structure to the piezoelectric substrate have a great influence on the performance of SAW strain sensors. The strain transfer rate will directly affect the sensitivity, and the transition zone will affect the accuracy of strain measurement. Therefore, a 3D model was established to analyze the strain transfer characteristics of the lithium niobate (LN) piezoelectric substrate (Figure 3) so as to maximize the sensitivity and accuracy of SAW strain sensors.

### 3.1. The Simplified 2D Model for Eigenfrequency Analysis

Since the displacement of RSAW in the y direction is zero [30] (as shown in Figure 1) and the length of IDTs is usually ten times longer than its width, the edge effect in IDTs can be ignored in the simulation. Thus, the 3D structure of a SAW device can be simplified as a 2D model [31]. Moreover, in general, the finger electrodes of IDTs are periodically arranged and alternatively biased by high and low voltages (in this paper, high voltage is 1 V and low voltage is 0 V), thus one period of the IDTs (see in Figure 2a) is sufficient to approximate the whole SAW resonator. 

As an anisotropic material, a different orientation of a cut crystal substrate will result in a different set of material properties, including the elastic matrix *C_E_*, coupling matrix *e* and relative dielectric constant matrix *ε_s_*, hence affecting the wave propagation characteristics. The selection of a unique crystal cut is defined by a set of Euler angles (α, β, γ) [27]. The Euler angle of 128° Y-cut LN is (0, −128°, 0) for the 2D model, and (0, −38°, 0) for the 3D model. Table 1, Table 2, Table 3 and Table 4 present the material parameters, structural parameters (see Figure 2b) and boundary conditions used in the simulation.

### 3.2. The Full-Sized 2D Model for the Total Displacement Field of RSAW and Frequency Domain Analysis

As shown in Figure 2b, the initial values for *N_t_* is 50, *N_r_* is 25 (in this paper, *N_t_* and *N_r_* on both sides are, respectively, equal), *L_tt_* is 20 μm, *L_rt_* is 5 μm and other structural parameters are shown in Table 3. The connection state of the RGs is open circuit, and each electrode of the RGs is set to be suspension potential. For the analysis of the total displacement field of RSAW, the finger electrodes of input IDTs are periodically arranged and alternatively biased by grounding and 1 V voltage. The finger electrodes of output IDTs are periodically arranged and alternatively biased by 0 C charge and grounding. For a frequency domain analysis, the finger electrodes of input IDTs are periodically arranged and alternatively biased by grounding and a termination power of 1 W. The finger electrodes of output IDTs are periodically arranged and alternatively biased by termination power of 0 W and grounding. See Table 5 for boundary conditions. 

### 3.3. Meshing

The energy of the RSAW is exponentially decaying into the material and is generally confined to a few wavelengths within the surface [30]. Therefore, relatively precise mesh cells are needed near the surface of the piezoelectric material under the electrodes. A mapping grid is used for subdivision for aluminum (Al) electrodes, LN piezoelectric substrate and PML layer. Among them, for the LN piezoelectric substrate, a reverse arithmetic sequence mesh is applied in z direction (i.e., in the thickness direction), where the number of elements is 15 and the element size ratio (i.e., the ratio of the maximum mesh cell area to the minimum mesh cell area) is 5. For the PML (perfect matching layer), eight layers of meshes are uniformly distributed in z direction. In x direction (i.e., the SAW propagation direction), meshing is carried out by controlling the maximum cell size, which is set as *λ*/*num*, where *num* is the number of meshes in each wavelength. Then, *num* is scanned parametrically near the eigenfrequency of RSAW with a proper step to acquire its optimized value. In order to display the results more intuitively, only the frequency range where RSAW is located is plotted in Figure 4, as is the case of the subsequent figures involving the results in the frequency domain. It can be seen that when *num* ≥ 8, the corresponding peaks of RSAW ultimately coincide, which indicates that the simulation accuracy has reached saturation when eight mesh cells are allocated to each wavelength. Therefore, *num* is set to 8 during the simulation of this model. As for the air domain, it has little effect on the simulation results, so the triangular mesh with conventional element size is used for subdivision.

### 3.4. A 3D Simulation Model for Strain Transfer Analysis

Since the strain distribution on the upper surface of LN is not uniform, a strain transition zone is built in order to obtain the strain distribution characteristics. The LN is divided into *N* strips with a wavelength of 20 μm (Figure 3). The larger layer at the bottom is Steel AISI 4340, which is used for generating strain, with a length of 64 mm, a width of 19.2 mm and a thickness of 0.8 mm. The cuboid above it is LN, with a length of 20 × *N* μm, a width of 0.5 × 20 × *N* μm and a thickness of *h_LN_*. See Table 1 and Table 2 for Steel AISI 4340 and LN piezoelectric material parameters, respectively. A force of 4 × 10^8^ N/m^2^ is applied in the x-direction of the S_1_-plane (Figure 3) and the fixed constrained boundary condition is used for the S_2_-plane, while all other planes are free.

## 4. Results and Discussion

### 4.1. Eigenfrequency Analysis

The admittance curve of the simplified 2D model (Figure 2a) under initial structural parameters (Table 3) is shown in Figure 5a and the inset figure shows a deformed mode when RSAW is excited. The resonant frequency is found to be around 195 MHz. By calculating the eigenfrequencies of the model with different electrode thicknesses and metallization rates, the resonant frequency varies: decreasing with the increase in *th_Al_* and *η* (shown in Figure 5b). This is mainly due to the mass increase with the increasing of electrode thickness and metallization rate, which is called the mass loading effect [33].

### 4.2. The Analysis of Total Displacement Field of RSAW

The eigenfrequency analysis of the full-sized 2D model under the initial structural parameters was performed, and the cloud map of the total displacement field of RSAW is shown in Figure 6. It shows an elliptical displacement of the RSAW. In addition, in the z direction (i.e., in the thickness direction), the energy of the RSAW is mainly confined to one wavelength, and in the x direction (i.e., the RSAW propagation direction), its energy is mainly concentrated between two sets of reflecting gratings, indicating that the RSAW is well excited.

### 4.3. Frequency Domain Analysis

According to the eigenfrequency analysis of the model shown in Figure 2a, the approximate resonant frequency *f_r_* of RSAW is 195.1 MHz under the initial structural parameters. Based on this, the frequency scanning of the model shown in Figure 2b under different structural parameters is carried out around the above frequency, and the scanning step is 0.1 MHz. As shown in Figure 7a, with the initial structural parameters (i.e., *N_t_* is 50), the corresponding frequency of the resonant peak in Rayleigh mode is 200.8 MHz, *IL* is 8.37 dB (i.e., *V_out_*/*V_in_* is 38.2%) and *Q* is 468.31. For a dual-port SAW resonator with the same material and structural parameters, the experimental results reported by Hongsheng Xu et al. [9] show that the frequency and *IL* are 195 MHz and 10 dB (i.e., *V_out_*/*V_in_* is 31.6%), respectively. Among them, the relative errors of frequency and *IL* are about 3% and 16.3%, respectively, and the absolute errors are 5.8 MHz and 1.63 dB (the corresponding *V_out_*/*V_in_* is only 6.6%), indicating that our simulation method is reasonable. The small difference may mainly originate from the following: first, the material parameters used by Hongsheng Xu et al. [9] are not exactly the same as those used in the simulation model; second, the coupling loss and electrode resistance loss are not considered in the simulation; third, the structural parameters of the actually manufactured dual-port SAW resonator are not exactly the same as those of the model used in the simulation due to the process errors.

The variation curves of *IL* and *Q* of RSAW with respect to the structural parameters of the resonator are shown in Figure 7 and Figure 8, where the insets show the resonant peaks of the Rayleigh mode under different structural parameters. As shown in Figure 7a, when the number of IDT finger pairs *N_t_* is lower than 60, *Q* of this resonator increases with *N_t_* and becomes to saturate with *N_t_* greater than 60. Meanwhile, the *N_t_* of 60 acts as an inflection point of *IL,* which declines at a small value of *N_t_* and increases thereafter. The loss of electric energy of IDTs due to an outward radiation of the SAW energy is represented by the acoustic radiation conductance $G_{0}$, which is calculated as [34]:(12)$$G_{0}\approx 8N_{t}K^{2}C_{T}f_{r}$$
(13)$$C_{T}=N_{t}WC_{0}$$
where *C*_0_ is the unit length capacitance of an IDT finger pair, *K* is the electromechanical coupling coefficient and *C_T_* is IDTs’ total electrostatic capacity with *N_t_* pairs of interfinger electrodes in parallel. According to Formula (12), *N_t_*^2^ is proportional to the acoustic radiation conductance; therefore, with the increase in the number of IDT finger pairs, the amount of electrical energy converted into mechanical energy increases, more energy is consequently emitted from the outward radiation and the *IL* becomes smaller. However, this analysis conflicts with the simulation results. The main reason may be that Formula (12) is derived based on the equivalent circuit model, which ignores the second-order effects as mentioned in the introduction section. It is because of these second-order effects that the *IL* increases rather than decreases when the number of IDT finger pairs exceeds a critical value.

The *Q* of SAW resonator can be expressed as:(14)$$Q=\frac{\pi L_{ef}}{\left(1-\left|\delta \right|\lambda \right)}$$
where *L_ef_* is the effective length of the cavity and *δ* is the reflection coefficient. It can be seen from Formula (14) that the higher *δ* is, the higher the *Q* of the resonator will be. *δ* can be expressed as:(15)$$\delta =tanhN_{r}\left|\frac{∆Z}{Z}\right|$$
where *Z* is the acoustic impedance in the free surface area between electrodes and the acoustic impedance in the electrode area is *Z_m_* = *Z* + Δ*Z*. Therefore, the performance of the resonator can be improved by increasing the number of RG pairs *N_r_*. However, with the increase in *N_r_*, the increasing of *Q* slows down and finally reaches saturation. The simulation results are shown in Figure 7b, which are in good agreement with the theoretical expectation. In addition, with the change of *N_r_*, *IL* shows the inverse trend as *Q*.

It can be seen from Figure 7c that with the increase in metallization rate *η*, *IL* first decreases and then increases, and shows a stable value at about *η* = 0.5. The *Q* increases first and then becomes stable after *η* = 0.6.

As shown in Figure 7d, with the increase in electrode thickness *th_Al_*, *IL* first decreases and then increases, and shows a minimum value at about *th_Al_* = 0.2 μm. In contrast, *Q* increases monotonically with *th_Al_*.

According to Equations (12) and (13), the increase in the acoustic aperture *W* will lead to the increase in the total electrostatic capacity of IDTs, thus increasing the acoustic radiation conductance and decreasing *IL*. On the other hand, if *W* is too large, the size of the resonator will become too big, and the loss will also increase. The variation curves of *Q* and *IL* with respect to *W* are shown in Figure 8a. It can be seen that with the increase in *W*, *Q* slowly increases, while *IL* first decreases and then remains nearly unchanged after *W* = 2.2 mm.

For a dual-port SAW resonator to generate standing waves, its *L_tt_* and *L_rt_* should satisfy the following relations:(16)$$L_{tt}=\frac{m}{2}\lambda$$
(17)$$L_{rt}=(n-\frac{1}{2})\frac{\lambda }{2}$$
where *m* and *n* are positive integers. As shown in Figure 8b,c, *Q* and *IL* fluctuate in a small range, where *Q* shows an overall downward trend and *IL* an upward trend with the increase in *L_tt_* and *L_rt_*; that is, in general, a smaller *L_tt_* and *L_rt_* are better to reduce the transmission loss and acquire a higher *Q*. Moreover, Figure 8b shows that the *IL* is relatively small when the distance between the input and output IDTs satisfies an odd multiple of the half-wavelength.

Since the energy of RSAW is mainly confined to the surface of the piezoelectric substrate, RSAW cannot be excited effectively when the substrate is too thin, and its energy will escape through the piezoelectric substrate, resulting in a large energy loss. As shown in Figure 8d, when the thickness of LN is less than 20 μm (i.e., one wavelength in this paper), the resonant peak of the Rayleigh mode has become insignificant. With the increase in LN thickness, *Q* gradually increases and finally reaches a stable value, while *IL* decreases first and then slightly increases.

### 4.4. The Steady State Analysis

After performing the steady state calculation on the model shown in Figure 3, the distribution of cloud map of the strain tensor component *S*_11_ is plotted in Figure 9a, which shows obvious strain gradients at both ends of the steel plate as well as the piezoelectric substrate. Therefore, the steel plate needs to be large enough to ensure that the piezoelectric substrate falls in its uniform strain region. In addition, to ensure the accuracy of the measurement of the strain sensor, the electrodes also need to be distributed in the uniform strain region of the piezoelectric substrate. Taking the strain transfer rate as a reference, the strain transfer characteristics of piezoelectric substrates under different thicknesses and lengths (i.e., Z and X directions) are analyzed, and the results are shown in Figure 9b,c. The expression of strain transfer rate σ
is:(18)$$\sigma =\frac{s_{LN}}{s_{st}}$$
where *s_st_* and *s_LN_* are the *S*_11_ component of strain tensors which belong to the upper surfaces of the steel plate and LN substrate, respectively. In addition, regions with default strain gradients ≤1% are defined as uniform strain regions (USRs).

As Figure 9b illustrates, with the increase in LN thickness, the strain transfer rate and the length of USR almost decrease linearly, indicating that the thicker the piezoelectric substrate is, the less favorable the strain transfer and the smaller the sensitivity of the strain sensor and the area available for electrode placement. Combined with the simulation results in Figure 8d, the thickness of LN piezoelectric substrate is selected as 40 μm, and then the length of LN is analyzed. As shown in Figure 9c, the strain transfer rate and the length of the USR increase largely linearly with the increase in the length of LN, which indicates that the larger the length of the piezoelectric substrate, the more conducive it will be to strain transfer, thus increasing the sensitivity of the strain sensor and the region available for placing electrodes. To avoid material waste, the size of the piezoelectric substrate should be specified in combination with the area required by the electrodes.

According to the above simulation results, the optimized structural parameters (see Table 6) of the dual-port SAW resonator based on LN is determined as follows:

The *Q* and *IL* of the dual-port SAW resonator used for strain sensing after structure optimization are 538.49 and 5.47 dB, respectively. Compared with before optimization, *Q* is increased by 15%, *IL* is decreased by 34.6% and strain transfer rate is increased by 2.4%, which fully demonstrate the significance of optimizing the structural parameters for SAW devices.

## 5. Conclusions

In this work, we propose a FEM simulation scheme for the dual-port SAW resonator, which is of great significance for the design and manufacture of various types of sensors based on the dual-port SAW resonator. The structure parameters, including *N_t_*, *N_r_*, *η*, *th_Al_*, *W*, *L_tt_*, *L_rt_* and *th_LN_*, are well optimized through eigenfrequency analysis, the frequency domain analysis as well as the steady state analysis. Compared with the experimental results reported in the literature, the relative errors of RSAW eigenfrequency and *IL* are about 3% and 16.3%, respectively, indicating that our simulation models are reasonable and accurate. Compared with the simulation results of the full-sized 2D model (Figure 2b) under the initial structural parameters, the dual-port SAW resonator *Q* increases by 15%, *IL* decreases by 34.6% and strain transfer rate increases by 2.4% after structural optimization, which indicates that this FEM simulation scheme can achieve an enhanced resonator performance.

## Figures and Tables

**Figure 1 materials-16-03269-f001:**
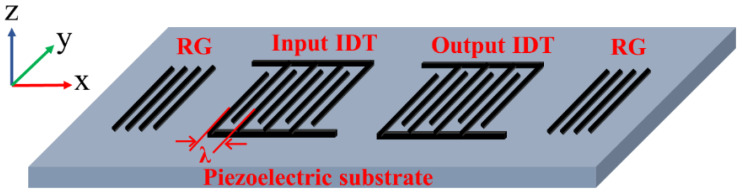
The schematic diagram of a dual-port SAW resonator.

**Figure 2 materials-16-03269-f002:**
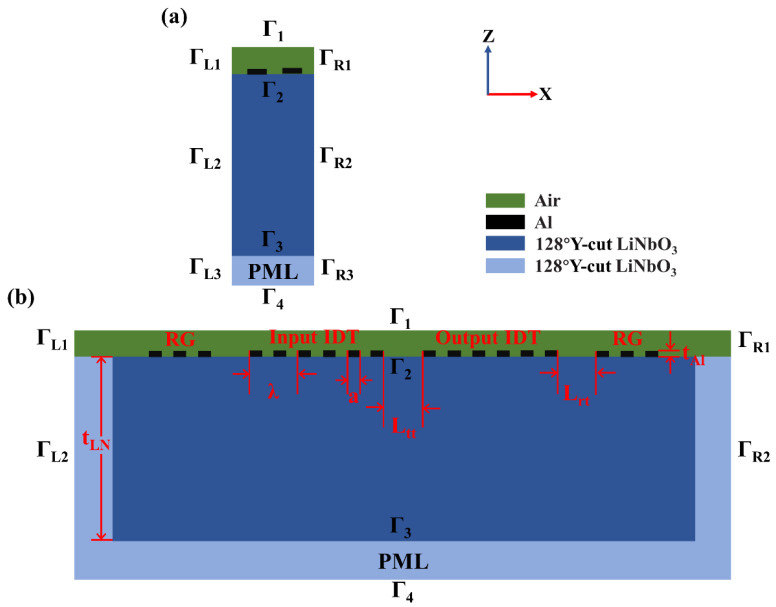
(**a**) The simplified 2D model with only one pair of electrodes (Г_i_ represents the boundaries). (**b**) The full-sized 2D model.

**Figure 3 materials-16-03269-f003:**
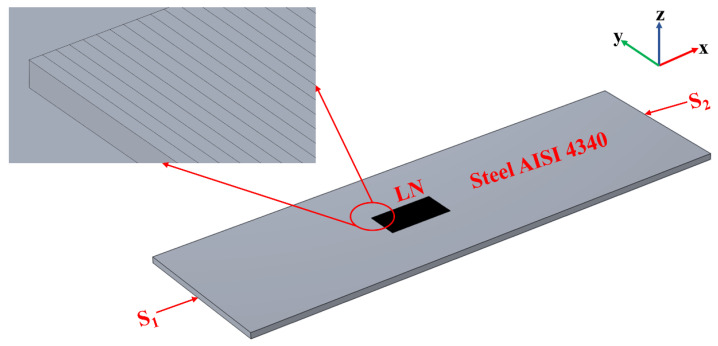
A 3D simulation model for strain transfer analysis.

**Figure 4 materials-16-03269-f004:**
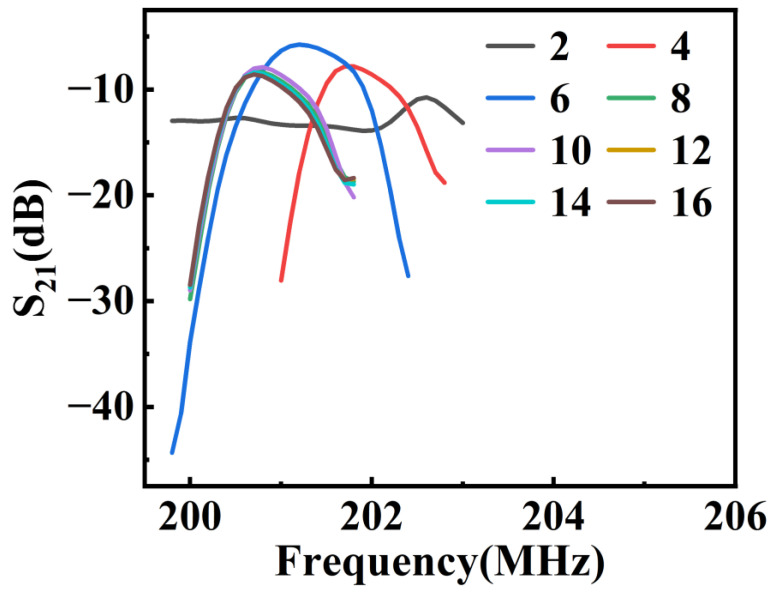
Frequency domain simulation with different *num* (number of meshes in each wavelength).

**Figure 5 materials-16-03269-f005:**
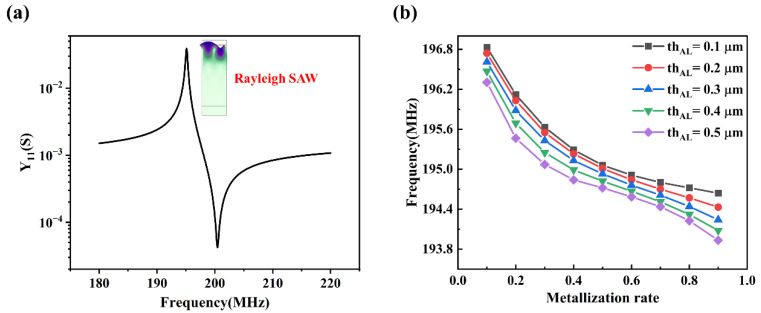
(**a**) The admittance curve of the simplified 2D model under initial structural parameters. (**b**) Variation curves of RSAW resonant frequencies with respect to metallization rates *η* and electrode thicknesses *th_Al_*.

**Figure 6 materials-16-03269-f006:**
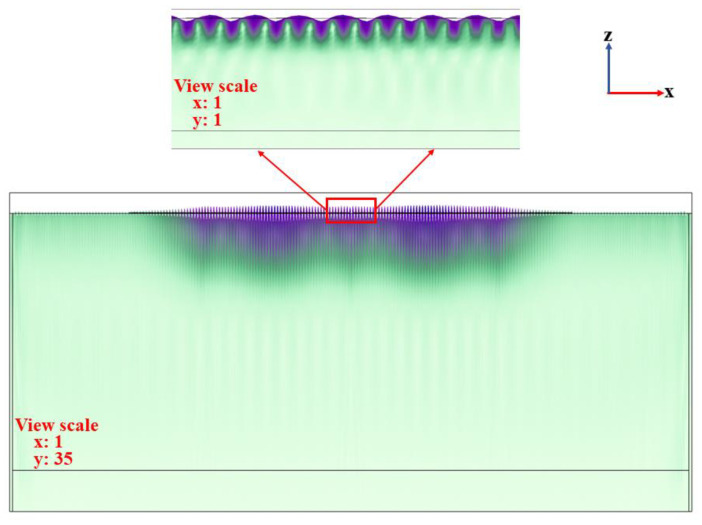
Distribution of the cloud map of the total displacement field of RSAW of the full-sized 2D model under initial structural parameters.

**Figure 7 materials-16-03269-f007:**
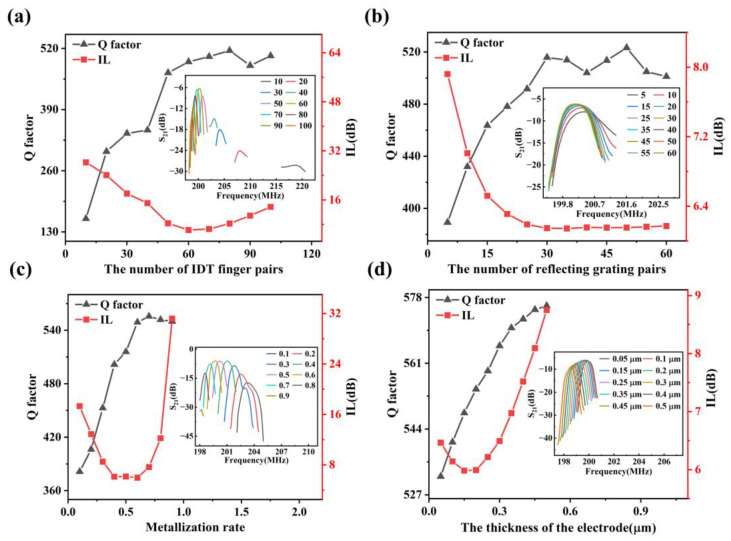
The variation curves of *IL* and *Q* of RSAW with respect to the structural parameters of the resonator, where the insets show the resonant peaks of the Rayleigh mode. (**a**) The number of IDT finger pairs. (**b**) The number of reflecting grating pairs. (**c**) Metallization rate. (**d**) The thickness of the electrode.

**Figure 8 materials-16-03269-f008:**
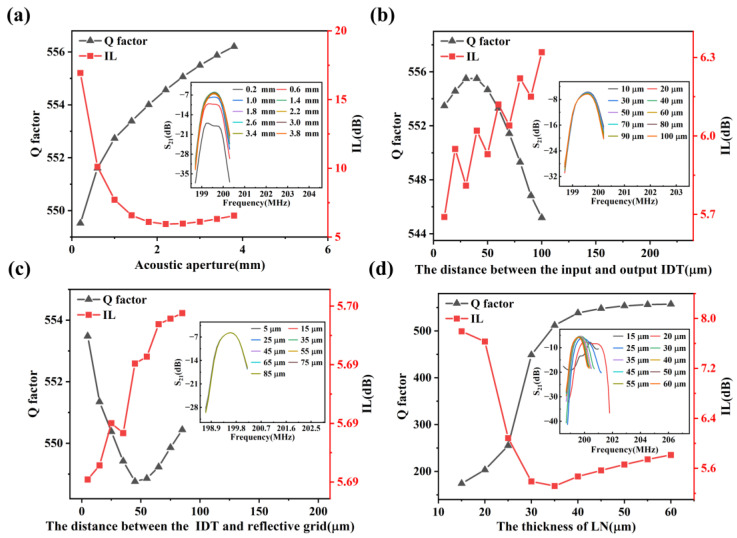
The variation curves of *IL* and *Q* of RSAW with respect to the structural parameters of the resonator, where the insets show the resonant peaks of the Rayleigh mode. (**a**) Acoustic aperture. (**b**) The distance between the input and output IDT. (**c**) The distance between the IDT and reflective grid. (**d**) The thickness of LN.

**Figure 9 materials-16-03269-f009:**
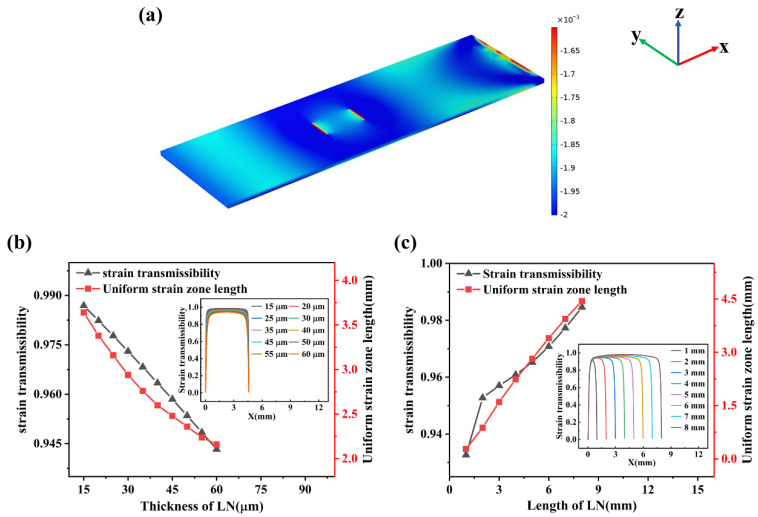
(**a**) The distribution cloud map of the strain tensor component *S*_11_; curves of strain transfer rate and uniform strain region length as a function of LN thickness (**b**) and LN length (**c**) where the inset shows the strain transmissibility on the upper surface of the piezoelectric substrate in X-direction.

**Table 1 materials-16-03269-t001:** Non-piezoelectric material parameters used in the simulation.

Material	Air	Al (Electrodes)	Steel AISI 4340
Density, *ρ* (kg/m^3^)	-	2700	7850
Relative dielectric constant, *ε*	1	1	-
Young’s modulus, *E* (Pa)	-	7 × 10^10^	2.05 × 10^11^
Poisson’s ratio, ν	-	0.33	0.28

**Table 2 materials-16-03269-t002:** Parameters of piezoelectric material used in the simulation.

Piezoelectric Material	LiNbO_3_
Density, *ρ* (kg/m^3^)	4700
Mechanical loss factor [32]	0.001
Dielectric loss factor [32]	0.005
Relative dielectric constant, *ε_s_*	$\left[\begin{array}{ccc} 43.6 & 0 & 0\\ 0 & 43.6 & 0\\ 0 & 0 & 29.16 \end{array}\right]$
Coupling matrix, *e* (C/m^2^)	$\left[\begin{array}{ccc} 0 & 0 & 0\\ -2.538 & 2.538 & 0\\ 0.194 & 0.194 & 1.309 \end{array}\begin{array}{ccc} 0 & 3.696 & -2.534\\ 3.695 & 0 & 0\\ 0 & 0 & 0 \end{array}\right]$
Elastic matrix, *C_E_* (1010 Pa)	$\left[\begin{array}{cccccc} 20.29 & 5.29 & 7.49 & 0.9 & 0 & 0\\ 5.29 & 20.3 & 7.49 & -0.9 & 0 & 0\\ 7.49 & 7.49 & 24.3 & 0 & 0 & 0\\ 0.9 & 0.9 & 0 & 6 & 0 & 0\\ 0 & 0 & 0 & 0 & 6 & 0.9\\ 0 & 0 & 0 & 0 & 0.9 & 7.5 \end{array}\right]$

**Table 3 materials-16-03269-t003:** Initialization of the simulation model.

Parameter	Value
The wavelength, *λ* (μm)	20
Metallization rate, *η* (i.e., *a*/(*λ*/2)	0.5
Electrode width, *a* (μm)	5
Acoustic aperture, *W* (mm)	2
Air field thickness, (μm)	4
Electrode thickness, *th_Al_* (μm)	0.16
128° Y-cut LN thickness, *th_LN_* (μm)	50
Perfect matching layer (PML) thickness, (μm)	8

**Table 4 materials-16-03269-t004:** Boundary conditions used in the 2D model of Figure 2a.

Boundary	Mechanical Conditions	Electrical Conditions
Г_1_	-	Zero charge
Г_2_, Г_3_	Free	Continuity
Г_4_	Fixed	Zero charge
Г_L2_, Г_L3_, Г_R2_, Г_R3_	Periodic boundary conditions	Periodic boundary conditions
Г_L1_, Г_R1_	-	Periodic boundary conditions

**Table 5 materials-16-03269-t005:** Boundary conditions used in the 2D model of Figure 2b.

Boundary	Mechanical Conditions	Electrical Conditions
Г_2_, Г_3_	Free	Continuity
Г_4_	Fixed	Zero charge
Г_L2_, Г_R2_	Free	Zero charge
Г_1_, Г_L1_, Г_R1_	-	Zero charge

**Table 6 materials-16-03269-t006:** Optimized structural parameters of a dual-port SAW resonator.

Parameter	Value	Parameter	Value
*λ* (μm)	20	*W* (mm)	2.2
*N_t_*	60	*L_tt_* (μm)	10
*N_r_*	30	*L_rt_* (μm)	5
*η*	0.6	*th_LN_* (μm)	40
*th_Al_* (μm)	0.2	the size of the LN (mm^2^)	7 × 3

## Data Availability

Not applicable.

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
