# Peer review of "Modulating the Performance of the SAW Strain Sensor Based on Dual-Port Resonator Using FEM Simulation"

_materials, 2023, doi:10.3390/ma16083269_

Round 1

Reviewer 1 Report

Simulation of a two-port resonator SAW strain gauge using COMSOL Multiphysics 5.6 finite element software provides interesting results. After optimizing the structure, the obtained resonator has higher Q and other improvements.

Mistakes:

The text after the equation should be without a paragraph.

Markings of individual parts (a-d) of the images is too large.

Table 3.  put on the new page

Errors in serial numbers of literature.

Rewrite the literature according to the requirements of the journal: Abbreviated Journal Name - italic, Volume  - italic.

In reference 33 the names of authors in Capital letters.

Reviewer 2 Report

1. Show the profile of total displacement of SAW amplitude over the delay line.

2. Why did not consider cryogenic temperature sensors

3. Authors should provide the measurement setup.

4.  why did not the authors calculate the quality factor (Q value)?

Reviewer 3 Report

The article deals with SAW strain sensor parametric analysis using finite elements.

- The English of the article has to be extensively rewied to ease the reviewing process. There are some words, sentences that are not very clear and/or well balanced.

- Figure 2 is not exploitable, especially 2(b), the authors should consider breaking it into 2 or 3 figures.

- The authors should change the title of the article : they are performing a parametric analysis not a structural optimization task since no optimizatio algorithm is involved so far

- The relative errors on IL are relatively large, how the authors justify the use of the simulation model for parametric analysis with such errors ?

- The "optimization" results are not confirmed by experiments. How the athors justify that ?

Round 2

Reviewer 3 Report

The article has been improved. The English should be reviewed by a native speaker or by the help of a dedicated software.

1-The authors should put forward in the abstract the novelty of their paper compared to existing analysis of the same kind in the literature

2- The authors mention that there are relative errors between their model and experimental results. The authors should mention why.

3- How the authors justify the use of model with errors (16.3% on IL for example) to perform structural optimization of SAW sensor ? To what extent the optimization results are correct ?
